# Cardiometabolic Phenotyping in Heart Failure: Differences between Patients with Reduced vs. Preserved Ejection Fraction

**DOI:** 10.3390/diagnostics13040790

**Published:** 2023-02-20

**Authors:** Alessio Balletti, Nicolò De Biase, Lavinia Del Punta, Francesco Filidei, Silvia Armenia, Filippo Masi, Valerio Di Fiore, Matteo Mazzola, Alessandra Bacca, Frank L. Dini, Stefano Taddei, Stefano Masi, Nicola Riccardo Pugliese

**Affiliations:** 1Department of Clinical and Experimental Medicine, University of Pisa, 56126 Pisa, Italy; 2Department of Pathology, Cardiology Division, University of Pisa, 56124 Pisa, Italy; 3Sant’Agostino Medical Center, 20127 Milano, Italy

**Keywords:** heart failure, inflammation, congestion, cardiometabolism

## Abstract

Aims. We explored multiple cardiometabolic patterns, including inflammatory and congestive pathways, in patients with heart failure (HF). Methods and Results. We enrolled 270 HF patients with reduced (<50%, HFrEF; *n* = 96) and preserved (≥50%, HFpEF; *n* = 174) ejection fraction. In HFpEF, glycated hemoglobin (Hb1Ac) seemed to be relevant in its relationship with inflammation as Hb1Ac positively correlated with high-sensitivity C-reactive protein (hs-CRP; Spearman’s rank correlation coefficient ρ = 0.180, *p* < 0.05). In HFrEF, we found a correlation between Hb1Ac and norepinephrine (ρ = 0.207, *p* < 0.05). In HFpEF, we found a positive correlation between Hb1Ac and congestion expressed as pulmonary B lines (ρ = 0.187, *p* < 0.05); the inverse correlation, although not significant, was found in HFrEF between Hb1Ac and N-terminal pro-B-type natriuretic peptide (ρ = 0.079) and between Hb1Ac and B lines (ρ = −0.051). In HFrEF, we found a positive correlation between E/e’ ratio and Hb1Ac (ρ = 0.203, *p* < 0.05) and a negative correlation between tricuspid annular systolic excursion (TAPSE)/echocardiographically measured systolic pulmonary artery pressure (sPAP) (TAPSE/sPAP ratio) (ρ = −0.205, *p* < 0.05) and Hb1Ac. In HFpEF, we found a negative correlation between TAPSE/sPAP ratio and uric acid (ρ = −0.216, *p* < 0.05). Conclusion. In HF patients, HFpEF and HFrEF phenotypes are characterized by different cardiometabolic indices related to distinct inflammatory and congestive pathways. Patients with HFpEF showed an important relationship between inflammatory and cardiometabolic parameters. Conversely, in HFrEF, there is a significant relationship between congestion and inflammation, while cardiometabolism appears not to influence inflammation, instead affecting sympathetic hyperactivation.

## 1. Introduction

Heart failure is a highly catabolic condition that can alter the metabolic balance of those affected. The gradual reduction of body weight, up to overt cachexia, characterizes chronic heart failure patients, especially those with reduced ejection fraction [1]. Patients with heart failure with reduced ejection fraction (HFrEF) in the early stages are characterized by a pro-atherogenic phenotype (dyslipidemia, overweight, obesity) in which weight loss could be beneficial to reduce the rate of ischemic complications [2]. Throughout their disease history, the atherosclerotic phenotype tends to overshadow in favor of a phenotype characterized by gradual weight loss, primarily based on malnutrition, which leads patients with HFrEF from overweight or obesity up to cachexia, especially in the final stages of the disease [1]. This concept is well explained by the “obesity paradox” [3]. Obesity is much more prevalent in patients with heart failure with preserved ejection fraction (HFpEF) compared to HFrEF, where over 80% of patients with HFpEF are overweight or obese. The obesity paradox is an interesting phenomenon reported in heart failure. Loss of body mass in obese patients improves left ventricular ejection fraction, NYHA class, and quality of life [4]. Paradoxically, it has been reported that a significant loss of body mass, even in overweight or obese patients, makes the prognosis worse and increases the risk of cachexia. In addition, patients with heart failure with BMI > 30 kg/m^2^ had greater survival than patients with normal weight [5]. More abundant adipose tissue could explain this in obese patients. They would thereby have further energetic reserves, which would be expected to drop the risk of cachexia. Thus, dysmetabolism in heart failure may play a major role.

To date, cardiometabolism remains largely unstudied. Therefore, the study of cardiometabolism is of considerable interest, especially concerning potential pathophysiological and therapeutic implications. Cardiometabolism is of complex evaluation. Indeed, there are no universally recognized, codified, and used parameters for the standard assessment of cardiometabolism to date. There are, however, widely recognized and utilized parameters of metabolic assessment; while not specifically for the heart failure patient, these may be of great interest in the near future. Of particular interest is the assessment of glucose (dys)metabolism in patients with heart failure (HF) [6,7,8]. Using validated metabolic indices could play a prominent role in improving our understanding of the pathophysiology of heart failure. Considering the gradual pro-atherogenic effect of hyperglycemia, glycated hemoglobin (Hb1Ac) is a predictive parameter of cardiovascular events in diabetic and nondiabetic patients [9]. Hb1Ac can also help assess microvascular engagement. Hb1Ac has been reported to correlate with inflammation in the general population [10]. To our knowledge, no studies have demonstrated the correlation between Hb1Ac and congestion or inflammation in heart failure. We hypothesized that Hb1Ac would have a different correlation with markers of congestion and inflammation in HFrEF and HFpEF. 

## 2. Methods

### 2.1. Study Population

We retrospectively evaluated 270 consecutive patients referred for a cardiovascular checkup at the University Hospital of Pisa (Italy) between September 2019 and September 2021. Physical examination and laboratory analyses were performed on the same day of the medical visit. Arterial hypertension was defined by at least two blood pressure (BP) recordings >140/90 mmHg or treatment with antihypertensive medications. Diabetes mellitus was defined by Hb1Ac ≥ 48 mmol/mol. HF diagnosis required at least two typical HF signs or symptoms: third heart sound, pulmonary rales, jugular venous distention, hepatomegaly, peripheral edema, or lung congestion on chest X-ray (performed in 143/270) or lung ultrasound (LUS, performed in 270/270). Patients were stratified into HFrEF (left ventricular ejection fraction, LVEF < 50%) and HFpEF (LVEF ≥ 50%, with the additional presence of N-terminal pro-B-type natriuretic peptide (NT-proBNP) > 125 pg/mL and relevant structural heart disease or diastolic dysfunction) [11]. No patient exercised regularly (more than once per week) at the time of the enrollment. 

### 2.2. Laboratory Analyses

Blood samples were collected on the same day of the medical visit. Patients were instructed to fast overnight and not to take any medications before blood sampling on the morning of the tests. Blood samples were drawn after a 30 min supine rest. The following analyses were performed: high-sensitivity C-reactive protein (hs-CRP), interleukin-6 (IL-6), NT-proBNP, Hb1Ac, and uric acid. Hs-CRP was measured with a quantitative turbidimetric test (TRUEchemie). NT-proBNP was measured with the ECLIA monoclonal assay using the Cobas e411 platform (Roche Diagnostics Italia, Monza, Italy). Plasma norepinephrine was evaluated using high-performance liquid chromatography with the electrochemical detector CLC 100 (Chromsystems, Munchen, Germany). Interleukin-6 was measured with an in vitro enzyme-linked immunosorbent assay for quantitative measurement (RayBiotech IL-6 ELISA kit).

### 2.3. Baseline Pulmonary Echography

All patients underwent an anterior thoracic echography examination (Hitachi Medical Systems LISENDO 880, Tokyo, Japan) to identify pulmonary B lines. Lung ultrasound (LUS) B-lines were measured in transverse orientation using an eight-region scan. In each region, B-lines were counted one by one if distinguishable; when they were confluent, the white screen percentage compared with the black screen below the pleural line was considered and divided by 10. Zero was defined as a complete absence of B-lines.

### 2.4. Baseline Echocardiography

All patients underwent a comprehensive transthoracic echocardiography examination (Hitachi Medical Systems LISENDO 880, Tokyo, Japan) according to the International Recommendations by the same expert physician (N.R.P.). LVEF was calculated using the Teicholz formula. Stroke volume (SV) was measured by multiplying the left ventricle (LV) outflow tract area by the LV outflow tract velocity-time integral measured by pulsed-wave Doppler. Systolic pulmonary artery pressure (sPAP) was measured from the peak tricuspid regurgitation velocity (TRV) with the simplified Bernoulli equation, adding the right atrial pressure estimated from imaging the inferior vena cava (IVC). 

Epicardial adipose tissue (EAT) thickness was identified as the echo-free space between the outer wall of the right ventricle (RV) and the visceral pericardium in the parasternal long-axis view at end-systole, averaged from two cardiac cycles, and with the best effort to be perpendicular to the aortic annulus, used as an anatomic landmark [12].

We measured left ventricular global longitudinal strain (LVGLS) from the apical long-axis view and two- and four-chamber views, ensuring a frame rate >50 Hz (2D strain analysis, TomTec Imaging Systems, Unterschleissheim, Germany). We reported the average LVGLS values from the three apical views at rest. We excluded poorly tracked segments, and patients were not analyzed if more than one segment per view was deemed unacceptable [13]. We measured the left atrial (LA) reservoir strain using the same software as the average strain in six segments in the four-chamber and two-chamber views, ensuring a frame rate >50 Hz. LA strain was measured using the QRS as the fiducial point. Speckle-tracking-echocardiography (STE)-derived measurements were reported as the average of three beats, and all measurements were performed offline by expert readers blinded to clinical and other instrumental data.

### 2.5. Statistical Analysis

We tested the data analytically for normal distribution using the Kolmogorov–Smirnov test. We reported continuous data as the mean ± standard deviation or median and interquartile range (IQR) for normally or skewed distributed variables, respectively. Continuous variables from two data sets were compared using Student’s *t*-test or Mann–Whitney U test for non-normal distributions. The relation between continuous variables was assessed using Spearman’s rank correlation coefficient ρ. Categorical variables were presented as percentages and were compared using the chi-square test or the Fisher exact test. All tests were two-sided, with a *p*-value of <0.05 considered significant. Data were analyzed with SPSS version 25.0 (IBM Corp., Armonk, NY, USA) and R 3.6.2 (R Foundation for Statistical Computing, Vienna, Austria).

## 3. Results

### 3.1. Study Population

The demographic and clinical characteristics of patients with HFpEF and HFrEF are shown in Table 1. Table 2 shows the distribution of blood analytes and total pulmonary B lines measured in the two cohorts of patients. In HFpEF, the median values of the inflammatory pathway were higher than in patients with HFrEF (hs-CRP 0.44 vs. 0.31 mg/dL, *p* < 0.05; IL-6 4.25 vs. 3.85 pg/mL). In patients with HFrEF, NT-proBNP was higher than in HFpEF patients (NT-proBNP 1428 vs. 885 pg/mL, *p* < 0.05), while norepinephrine was higher in HFpEF (norepinephrine 354 vs. 297 pg/mL, *p* < 0.05). The median values of Hb1Ac were similar between the two groups. Most cardioactive drugs were significantly more common in HFrEF than in HFpEF.

### 3.2. Association of Cardiometabolic Profile with Inflammation and Congestion Indices (Figure 1)

In HFpEF patients, a positive correlation between inflammatory analytes (hs-CRP, IL-6) and glycated hemoglobin was found. In HFpEF, a positive correlation between congestion indices (NT-proBNP, pulmonary B lines) and glycated hemoglobin was also identified. In HFpEF, there was a positive correlation between uric acid and glycated hemoglobin. Opposite results were documented in patients with HFrEF: a negative correlation was identified between inflammatory analytes (hs-CRP, IL-6) and glycated hemoglobin. In HFrEF, Hb1Ac correlated inversely with congestion parameters (NT-proBNP, pulmonary B lines) and uric acid. In HFrEF, Hb1Ac was positively associated with norepinephrine, while the opposite result was found in HFpEF. In HFpEF patients, a stronger correlation between plasma norepinephrine and interleukin-6 was found than in HFrEF patients. In HFrEF patients, a stronger correlation between norepinephrine and NT-proBNP was identified compared to HFpEF patients. Conversely, in patients with HFpEF, a stronger positive correlation between norepinephrine and IL-6 was found compared to HFrEF patients. Both in HFpEF and HFrEF, uric acid positively correlates with norepinephrine. 

### 3.3. Association of Cardiometabolic Profile with Diastolic Function and Right Ventriculo-Arterial Coupling

In HFrEF, there was a significant positive correlation between the ratio of early diastolic mitral inflow velocity to early diastolic mitral annulus velocity (E/e’ ratio) and Hb1Ac (ρ = 0.253, *p* = 0.02). In HFpEF, the same positive correlation was found between E/e’ ratio and Hb1Ac (ρ = 0.228; *p* = 0.04). In HFpEF, we found a statistically significant inverse correlation between the ratio of tricuspid annular plane systolic excursion/systolic pulmonary artery pressure (TAPSE/sPAP ratio) and uric acid (ρ = −0.246, *p* = 0.02). In HFrEF, there was a significant inverse correlation between TAPSE/sPAP ratio and Hb1Ac (correlation ρ = −0.235, *p* = 0.03). 

## 4. Discussion 

Our study investigated the relationship between cardiometabolism, congestion, and inflammation in heart failure (Figure 2). We evaluated the correlation between cardiometabolism expressed by glycated hemoglobin and uric acid and the vicious cycle of inflammation, congestion, and sympathetic hyperactivity [14,15]. Cardiometabolism is an indispensable parameter for assessing the patient with HFrEF and HFpEF. Despite the importance of cardiometabolism, the latter is not routinely studied in patients with heart failure. Nevertheless, the evidence giving a prominent role to the assessment of metabolism in patients with heart failure is recent and not yet established. No less important is that, to date, there are no validated indices of cardiometabolic assessment. Growing evidence demonstrates the importance of assessing cardiometabolism in the pathophysiology of heart failure. The exact role of cardiometabolism in patients with heart failure as well as in the genesis and prognosis of the disease remains to be investigated.

### 4.1. Correlation of Cardiometabolism with Inflammation and Congestion

Metabolic derangements play essential roles in heart failure initiation and progression. In our cohort of patients with HFpEF, Hb1Ac correlated to hs-CRP. Hb1Ac also correlated to pulmonary B lines. Interestingly, in HFrEF, we found inverse correlations between Hb1Ac and inflammation and congestion. Metabolic alterations in heart failure are complex, partially due to the various disease etiologies of the syndrome [19]. The stage and severity of heart failure also play important roles in determining the oxidative capacity of the heart. Overdrive of the sympathetic nervous system and insulin resistance contribute significantly to rises in circulating free fatty acid (FFA) levels and consequent increases in FFA oxidation [20]. For these reasons, glucose uptake and utilization may be suppressed. For that reason, we have used glycated hemoglobin (Hb1Ac) as an indirect and reliable index of cardiometabolism. We also added uric acid to this biochemical evaluation of cardiometabolism. We found a correlation between uric acid and norepinephrine in HFpEF. The adrenergic activation in HFpEF may be explained by the lack of optimal medical therapy supported by scientific evidence contributing to the failure to control the adrenergic overdrive [21]. In HFrEF, we did not find a significant correlation between uric acid and norepinephrine. At the same time, we found an interesting correlation between Hb1Ac and norepinephrine in HFrEF. Metabolism is an important parameter to be evaluated in heart failure. In HFpEF, the increased body mass index (BMI) tends to decrease less over time than in patients with HFrEF [3]. Indeed, in patients with HFrEF, the BMI could significantly decrease in the course of the disease, up to cachexia. That is why cardiac cachexia is more prevalent in HFrEF patients [4]. We hypothesized that because of the positive correlation between Hb1Ac and norepinephrine in HFrEF and the concomitant lack of correlation between Hb1Ac and inflammation, Hb1ac might be an index of disease severity in HFrEF. Sympathetic overactivation may exert a detrimental effect on the myocardium and be more present in advanced stages of HFrEF. At the same time, regarding inflammation, in patients with low BMI, which corresponds to lower muscle mass and fat mass, the myocardium might metabolize less glucose due to lower demand; this might explain the lack of correlation between Hb1Ac, inflammation, and congestion. In contrast, in patients with HFpEF, we hypothesized that glycated hemoglobin might be a helpful index to understand the pathophysiology of the disease better. It is well known that patients with HFpEF are also characterized by marked inflammatory activation [14]. The correlation between Hb1Ac and inflammation may further confirm the role of glucotoxicity as responsible for the production of reactive oxygen species (ROS), which are factors of myocardial damage [22]. In patients with HFrEF, we did not find a correlation between Hb1Ac and inflammation, probably due to different pathogenetic mechanisms between the two heart failure phenotypes. HFpEF has been demonstrated to be a “metabolic/inflammatory” phenotype of heart failure, mainly characterized by obesity, increased epicardial adipose tissue [12], and microvascular rarefaction, among others [23]. Hyperuricemia is a surrogate index of an increased level of xanthine oxidase. Xanthine oxidase and xanthine dehydrogenase were highly expressed in the myocardium of patients with dilated cardiomyopathy [24]. Xanthine oxidase could provide a source of reactive oxygen species sufficient to activate cardiac afferents during ischemia, and its inhibition significantly reduced the responses of cardiac sympathetic afferents during ischemia [25]. Therefore, xanthine oxidase may be closely linked to sympathetic nerve fiber activity and could have a detrimental effect on cardiac function and the prognosis of heart failure [26]. Both xanthine oxidase and sympathetic overactivation could produce reactive oxygen species, which might be closely related to the pathophysiology of heart failure [27]. In patients with HFrEF, we found a nonsignificant correlation between uric acid and norepinephrine, probably because beta blockers taken under optimal therapy may protect the myocardium from myocyte apoptosis induced by beta-adrenergic receptor activation. In contrast, adrenergic overdrive does not appear well controlled in patients with HFpEF, even in those on beta-blocker therapy. We hypothesized that cardiometabolism in patients with HFpEF might be more important than sympathetic overdrive in the pathophysiology of the disease.

### 4.2. Correlation of Inflammation with Congestion and Adrenergic Activation

HFrEF is a clinical syndrome characterized by signs and symptoms of congestion in the context of a reduced left ventricular systolic function [2]. In our cohort of chronic patients on optimal medical therapy, congestion (testified by NT-proBNP) was correlated to hs-CRP; residual adrenergic activation (norepinephrine) was correlated to inflammation, expressed by hs-CRP. HFpEF is characterized by an elevation of left ventricle filling pressure with concomitant diastolic dysfunction, structural abnormalities, and elevation of natriuretic peptides [13,28,29,30]. In our cohort of HFpEF patients, congestion seemed correlated to both adrenergic drive expressed by norepinephrine and inflammation (hs-CRP); interestingly, there was also an interplay between interleukin-6 and norepinephrine which is unrelated to the state of congestion. Such differences between HFrEF and HFpEF may relate to the comprehensive spectrum of anti-neurohormonal treatments available in the context of systolic dysfunction, in which a residual state of congestion could be attributed to an underlying inflammatory state [31]. On the other hand, in HFpEF, which is characterized by diastolic dysfunction and elevated filling pressures, the more limited therapeutic strategies reflect a greater amount of adrenergic escape [32]. It should also be noted that many comorbidities dominate the pathophysiology of HFpEF with a significant inflammatory stimulus, from diabetes to hypertension to chronic obstructive pulmonary disease (COPD) [33]. HFrEF, instead, is generally due to ischemic damage or primitive or secondary dilated cardiomyopathy: such processes may involve inflammation acutely (e.g., in myocarditis or myocardial infarction), but this mechanism can hardly be considered as the main one in the chronic course of the disease. In turn, pulmonary, hepatic, renal, and peripheral congestion caused by systolic dysfunction may result in a reactive inflammatory response in tissues. In this perspective, inflammation may be seen as a cause of HFpEF and as a consequence of HFrEF [14].

### 4.3. Correlation of Cardiometabolism with Diastolic Dysfunction and Right Ventricular Arterial Coupling

Both HFrEF and HFpEF can be characterized by varying degrees of diastolic dysfunction. E/e’ ratio documents a diastolic dysfunction characterized by high ventricular filling pressure by tissue Doppler imaging (TDI). It is now known that diastolic dysfunction characterized by increased E/e’ is of considerable importance in the prognosis of patients with HFrEF [34] and HFpEF [35]. Advanced diastolic dysfunction is a possible feature of advanced heart failure with reduced ejection fraction. If we consider Hb1Ac an index of disease severity in HFrEF, it is fair to consider that metabolic and hemodynamic changes go hand in hand. Therefore, we hypothesized that the positive correlation between diastolic dysfunction and Hb1Ac in HFrEF may identify a severe phenotype.

In both HFrEF and HFpEF, there are negative correlations between cardiometabolism and right ventricular arterial coupling parameters. The TAPSE/sPAP ratio has been identified as a good index of right heart function as well as prognosis in patients with heart failure [36]. In particular, a reduction in the TAPSE/sPAP ratio appears to be related to worse survival in patients with heart failure [37]. In patients with HFrEF, the negative correlation between TAPSE/sPAP and Hb1Ac confirms the hypothesis of Hb1Ac as an index of disease severity. From what has emerged so far, the exact role Hb1Ac plays in the pathophysiology of disease in HFrEF is challenging. However, it is interesting to note the close relationship that Hb1Ac maintains with the main parameters of poor prognosis. We hypothesized that although the role of glucose metabolism in HFrEF needs to be further investigated, Hb1Ac may be a good potential parameter of disease severity. 

In HFpEF, a negative correlation between TAPSE/sPAP ratio and uric acid confirms the association between hyperuricemia and worse prognosis [38]. It is interesting to note that, once again, cardiometabolism is a parameter that goes hand in hand with those echocardiographic parameters suggestive of poor prognosis.

### 4.4. Clinical Perspectives

Routine and repeated assessment of glycated hemoglobin over time, even in nondiabetic heart failure patients, could be helpful in the integrated assessment of disease severity in HFrEF. In patients with HFpEF, the evaluation of glycated hemoglobin and uric acid combined with the measurement of hs-CRP could be helpful to the clinician in assessing cardiometabolism. Indeed, the latter constitutes an exciting feature of patients with HFpEF that could be routinely implemented in HF laboratories. The study of cardiometabolism would justify the development and use of old and new drugs active on glucose and uric acid metabolic pathways. For example, SGLT2i, initially approved only for the treatment of diabetes, can improve the prognosis of patients with HF, irrespective of LVEF [39], and increasing evidence indicates that this benefit could derive from their anti-inflammatory properties and more favorable metabolic profile [40,41,42].

### 4.5. Limitations

This is a single-center study from a tertiary referral center: it has inherent flaws related to selection and referral bias and the absence of a validation cohort. As an observational study, causality cannot be deduced based on these data. We classified all patients with LVEF < 50% as HFrEF because only 31 cases had mildly reduced LVEF (41–49%). We performed a sensitivity analysis revealing a similar distribution of demographics, cardiometabolic profile, and indices of inflammation and congestion between patients with LVEF ≤ 40% and HFmrEF. We assessed cardiometabolism using only an index related to glucose and uric acid metabolism. There are other metabolic assessment parameters that we did not take into account that would be interesting to evaluate in future studies. We defined diabetes mellitus only by Hb1Ac ≥ 48 mmol/mol; thus, we might not have identified all the participants with diabetes. Different protocols have been proposed for performing LUS, and our findings may have been different if we had used another method. B-lines are not specific for pulmonary congestion and may also reflect parenchymal lung disease, although we excluded moderate-to-severe lung disease with spirometry. 

## 5. Conclusions

Despite the limitations of using correlations, the data suggest the importance of cardiometabolism in patients with heart failure. In patients with HFrEF, using direct and indirect markers of cardiometabolism, in addition to those we have used, could help assess disease severity. In patients with HFpEF, cardiometabolism seems to play a more critical role than adrenergic overdrive in disease pathophysiology. The significant presence of inflammatory overdrive could be one of the causes of the disease, and metabolic dysregulation could be its consequence or epiphenomenon.

## Figures and Tables

**Figure 1 diagnostics-13-00790-f001:**
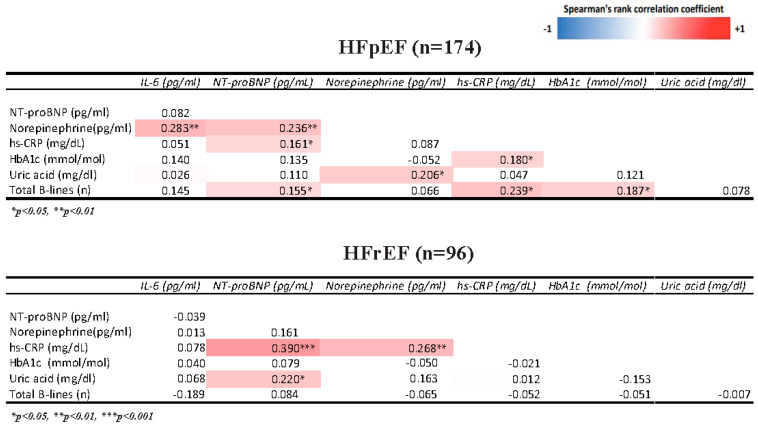
Univariate correlation matrix for different markers of cardiometabolism, inflammation, and congestion using Spearman’s Rank correlation. Red shading indicates positive correlations, and blue shading indicates inverse correlations. White boxes are nonsignificant (*p* > 0.05). HbA1c: glycated hemoglobin; hs-CRP: high-sensitivity C-reactive protein.

**Figure 2 diagnostics-13-00790-f002:**
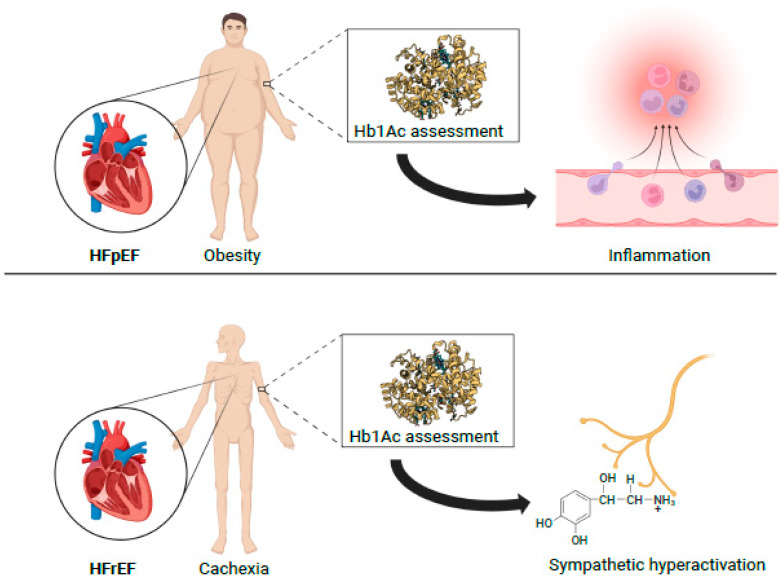
Cardiometabolic phenotyping using glycated hemoglobin. Hb1Ac as a tool to understand the pathophysiology of heart failure with preserved ejection fraction. Hb1Ac in heart failure with reduced ejection fraction as a tool to assess disease severity. HbA1c: glycated hemoglobyn; HFpEF: heart failure with preserved ejection fraction; HFrEF: heart failure with reduced ejection fraction. Indeed, it has been demonstrated from a molecular and cellular perspective that metabolic dysregulation is essential in the pathophysiology of heart failure [16]. The myocardium is a metabolically very active tissue that can be affected by glucose and lipid metabolic alterations. In fact, under normal conditions, the myocardium uses predominantly fatty acids as a substrate to produce energy [17]. In heart failure, the shift from fatty acid utilization to glucose utilization as an energy source is well known [18]. Our study expands the usefulness of assessing cardiometabolism as an index of severity in HFrEF and a potential therapeutic target in HFpEF.

**Table 1 diagnostics-13-00790-t001:** Population characteristics.

Variable	HFrEF (*n* = 96)	HFpEF (*n* = 174)	*p*-Value
**Demographics**			
Age, years	73 ± 11.4	75 ± 11.6	0.1
Male	77 (80.2)	97 (55.7)	0.1
BMI, Kg/m^2^	25.5 (23.2–28.9)	27.5 (25.4–30.2)	**<0.001**
Waist circumference, cm	80 (71–91)	95 (91–106)	**<0.001**
Obesity	25 (26)	44 (25.3)	0.8
Smoker	21 (21.9)	28 (16)	0.2
**Clinical evaluation**			
NYHA I	27 (28.1)	45 (25.8)	**<0.001**
NYHA II	49 (51)	83 (47.7)	0.1
NYHA III	14 (14.5)	41 (23.5)	0.1
Arterial hypertension	57 (59.3)	126 (72.4)	**0.028**
Dyslipidaemia ^#^	60 (62.5)	94 (54)	0.1
Stroke/TIA	8 (8.3)	24 (13.7)	0.2
Diabetes mellitus	29 (30.2)	56 (32.1)	0.7
CKD	50 (42)	43(38)	0.5
CAD	38 (39.5)	49 (28.1)	**0.05**
Previous PCI/CABG	33 (34.4)	43 (24.7)	0.09
Previous MI	29 (30.2)	27 (15.5)	**0.004**
ICD	25 (26)	22 (12.6)	**0.006**
**Therapy**			
Beta-blocker	82 (70)	102 (64)	**<0.0001**
DHP CCB	33 (16)	47 (25)	0.08
Non-DHP CCB	0	4 (2)	0.1
ACEi or ARB	87 (71)	132 (70)	0.1
MRA	83 (49)	71 (38)	**<0.0001**
Digoxin	21 (10)	13 (7)	**0.01**
ARNI	45 (22)	0	**<0.0001**
Statins	87 (65)	90 (49)	**<0.0001**
Thiazides	28 (14)	20 (10)	**0.05**
Furosemide	89 (70)	20 (11)	**0.001**
Insulin	7 (7)	16 (9)	**0.001**
SGLT2i	29 (30)	22 (13)	**0.001**
Other oral hypoglycaemic drugs	21 (22)	33 (19)	0.5

Values are mean ± standard deviation and *n* (%). ^#^ total cholesterol ≥ 200 mg/dL or LDL ≥ 130 mg/dL or on lipid-lowering therapy. ACEi: angiotensin-converting-enzyme inhibitor; ARB: angiotensin receptor blocker; ARNI: angiotensin receptor neprilysin inhibitor; BMI: body mass index; CABG: coronary artery bypass graft; CAD: coronary artery disease; CKD: chronic kidney disease (eGFR < 60 mL/min/1.73 m^2^); DHP CCB: dihydropyridine calcium channel blocker; eGFR: estimated glomerular filtration rate; HFpEF: heart failure with preserved ejection fraction; HFrEF: heart failure with reduced ejection fraction; ICD: implantable cardioverter defibrillator; MI: myocardial infarction; MRA: mineralocorticoid receptor antagonist; PCI: percutaneous coronary intervention; SGLT2i: sodium-glucose transport protein 2 inhibitor; TIA: transient ischemic attack.

**Table 2 diagnostics-13-00790-t002:** Markers of cardiometabolism, inflammation, and congestion in heart failure.

Variable	HFrEF (*n* = 96)	HFpEF (*n* = 174)	*p*-Value
HbA1c, mmol/mol	41 (37.8–45)	40 (37–44)	0.5
Norepinephrine, pg/mL	297 (153–403)	354 (255–523)	**0.03**
Uric acid, mg/dL	6.3 (4.8–7.5)	8.1 (5.9–9.9)	**0.01**
hs-CRP, mg/dL	0.31 (0.21–0.57)	0.44 (0.31–0.63)	**0.03**
NT-pro BNP, pg/ml	1428 (506–3155)	885 (442–1163)	**0.01**
Interleukin-6, pg/mL	3.85 (1.5–9)	4.25 (1.6–8.42)	**0.6**
Total pulmonary B lines	2 (0–8)	2 (0–7)	0.9

Values are median (25th quartile, 75th quartile). Hb1Ac: glycated hemoglobin; hs-CRP: high-sensitivity C-reactive protein; NT-proBNP: N-terminal prohormone of brain natriuretic peptide.

## Data Availability

Data available on request.

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
