# Peer review of "Cardiometabolic Phenotyping in Heart Failure: Differences between Patients with Reduced vs. Preserved Ejection Fraction"

_diagnostics, 2023, doi:10.3390/diagnostics13040790_

Round 1

Reviewer 1 Report

Strongly suggest for publication.

Author Response

We thank the Reviewer for her/his report.

Reviewer 2 Report

Manuscript ID: diagnostics-2194733

Title: "Cardiometabolic phenotyping in heart failure: differences between patients with reduced vs preserved ejection fraction".

Authors: Alessio Balletti et al.

The authors of the present manuscript tried to explore the potential associations of various cardiometabolic factors with HFrEF and HFpEF.

The following points should be considered.

Comments:

1.     Considering the low values of coefficient, r, and the non-significant association, I would suggest rephrasing or omitting the relevant statements from the abstract (e.g., "In HFrEF, an inverse correlation, although not significant, was found between Hb1Ac and hs-CRP (r=-0.050) and between Hb1Ac and IL-6 (r=0.040).")

2.     Please clarify further the aim(s) of the present study (primary and secondary).

3.     Please add the relevant references corresponding to HFrEF and HFpEF definitions.

4.     According to the authors, 31 cases had Heart Failure with mildly reduced ejection fraction (HFmrEF). Therefore, the actual number of participants with HFrEF is 65. I would also suggest examining these conditions separately and calculating the necessary sample size.

5.     Please define the type of blood samples and whether they were randomly collected or after fasting.

6.     Please define if the same physicians performed the echocardiography examination.

7.     Please define how normality was assessed. Moreover, according to the authors, "We used the Pearson's correlation coefficient r to assess the relationship between the two variables". Were all the variables normally distributed (i.e., since the authors used only the Pearson correlation coefficient to explore the associations)?

8.     Did the authors include SGLT2 drugs in the "Oral hypoglycaemic drugs" variable? Considering the significant role of SGLT2 on HF treatment, I would suggest presenting the SGLT2 data separately.

9.     Moreover, it would be useful if the authors could also assess the physical activity of the participants.

10.  Could the authors clarify further the "congestive pathways in heart failure" term?

11.  It would be important if the authors could perform logistic regression for HFpEF and HFrEF, exploring which variables are independently associated with these conditions after adjusting for potential confounding factors.

12.  Please comment on whether the COVID-19 pandemic may potentially impact patients who attended the clinic during the study period.

13.  According to the authors, "Diabetes mellitus was defined by Hb1Ac ≥48 mmol/mol.". By using only the A1c criterion, the authors might not have identified all the participants with diabetes. Please acknowledge the above limitation.

Minor comments:

1.     Please define the abbreviations the first time used in the abstract and main text of the manuscript (e.g., HbA1c).

Reviewer 3 Report

In my opinion, the article is well written, the research methods are modern, the conclusions are justified. It should be noted the undoubted practical importance of this work for the therapy and diagnosis of heart failure. In my opinion, the article could in principle be published in this form, but please pay attention to the comments on page 3 in Italian.

Author Response

We thank the Reviewer for her/his report. We deleted the comments from the manuscript.

Round 2

Reviewer 2 Report

Manuscript ID: diagnostics-2194733 Revised

Title: "Cardiometabolic phenotyping in heart failure: differences be-2 tween patients with reduced vs preserved ejection fraction".

Authors: Alessio Balletti et al.

The authors tried to respond to my comments and make the appropriate changes. As a result, the revised manuscript has been improved. However, the following points merit consideration:

Comments:

1.     According to the revised manuscript, it should be noted that the number of participants receiving SGLT2i differs significantly between the two groups. Moreover, considering that accumulating evidence has highlighted the anti-inflammatory properties of SGLT2i and their potential association with uric acid, the investigators should explore the potential impact of SGLT2i in the study findings. Moreover, SGLT2i should be discussed in the discussion section.

(indicatively: J Am Coll Cardiol. 2020 Feb 4;75(4):422-434; Front. Cardiovasc. Med. 9:1008922. doi: 10.3389/fcvm.2022.1008922; Cell. Mol. Life Sci. 79, 273 (2022). https://doi.org/10.1007/s00018-022-04289-z)

2.     Regarding my previous comment #11: "It would be important if the authors could perform logistic regression for HFpEF and HFrEF, exploring which variables are independently associated with these conditions after adjusting for potential confounding factors.". Did the authors include the SGLT2i in their analyses (considering the above points)? It would be helpful to include and discuss the findings in the manuscript.

Round 3

Reviewer 2 Report

Manuscript ID: diagnostics-2194733 Revised version 2

There are no further comments.